Key auxin response factor (ARF) genes constraining wheat tillering of mutant dmc

Li Junchang 1
Jiang Yumei 1
Zhang Jing 1
Ni Yongjing 2
Jiao Zhixin 1
Li Huijuan 1
Wang Ting 1
Zhang Peipei 1
Guo Wenlong 1
Li Lei lilei@henau.edu.cn 1
Liu Hongjie 2
Zhang Hairong 3
Li Qiaoyun 1
Niu Jishan jsniu@henau.edu.cn 1
1 National Centre of Engineering and Technological Research for Wheat/National Key Laboratory of Wheat and Maize Crop Science, Henan Agricultural University , Zhengzhou , Henan , China
2 Shangqiu Academy of Agricultural and Forestry Sciences , Shangqiu , Henan , China
3 College of Life Sciences, Henan Agricultural University , Zhengzhou , Henan , China
Huang Shanjin
Electronic publication date: 2021 Sep 21
Publication date: 2021
Volume: 9
Electronic Location ID: e12221
Received 2021 Jun 26; Accepted 2021 Sep 6
Copyright: ©2021 Li et al.
Copyright year: 2021
Copyright holder: Li et al.
License: This is an open access article distributed under the terms of the Creative Commons Attribution License, which permits unrestricted use, distribution, reproduction and adaptation in any medium and for any purpose provided that it is properly attributed. For attribution, the original author(s), title, publication source (PeerJ) and either DOI or URL of the article must be cited.
License URL: https://creativecommons.org/licenses/by/4.0/

Keywords: Wheat (Triticum aestivum L.), Tillering, Auxin response factor, Expression profiles, IAA

Funding: National Key R & D Program of China 2017YFD0301101 The Science and Technology Project in Henan Province 212102110060 The open project fund of National Key Laboratory of Wheat and Maize Crop Science, Henan Agricultural University (2020) This study was supported by the National Key R & D Program of China (2017YFD0301101), the Science and Technology Project in Henan Province (212102110060) and the open project fund of National Key Laboratory of Wheat and Maize Crop Science, Henan Agricultural University (2020). The funders had no role in study design, data collection and analysis, decision to publish, or preparation of the manuscript.

==============================
Tillering ability is a key agronomy trait for wheat (Triticum aestivum L.) production. Studies on a dwarf monoculm wheat mutant (dmc) showed that ARF11 played an important role in tillering of wheat. In this study, a total of 67 ARF family members were identified and clustered to two main classes with four subgroups based on their protein structures. The promoter regions of T. aestivum ARF (TaARF) genes contain a large number of cis-acting elements closely related to plant growth and development, and hormone response. The segmental duplication events occurred commonly and played a major role in the expansion of TaARFs. The gene collinearity degrees of the ARFs between wheat and other grasses, rice and maize, were significantly high. The evolution distances among TaARFs determine their expression profiles, such as homoeologous genes have similar expression profiles, like TaARF4-3A-1, TaARF4-3A-2 and their homoeologous genes. The expression profiles of TaARFs in various tissues or organs indicated TaARF3, TaARF4, TaARF9 and TaARF22 and their homoeologous genes played basic roles during wheat development. TaARF4, TaARF9, TaARF12, TaARF15, TaARF17, TaARF21, TaARF25 and their homoeologous genes probably played basic roles in tiller development. qRT-PCR analyses of 20 representative TaARF genes revealed that the abnormal expressions of TaARF11 and TaARF14 were major causes constraining the tillering of dmc. Indole-3-acetic acid (IAA) contents in dmc were significantly less than that in Guomai 301 at key tillering stages. Exogenous IAA application significantly promoted wheat tillering, and affected the transcriptions of TaARFs. These data suggested that TaARFs as well as IAA signaling were involved in controlling wheat tillering. This study provided valuable clues for functional characterization of ARFs in wheat.

Introduction

Auxin response factors (ARFs) belong to a subfamily of plant B3 superfamily, and they are a kind of plant-specific transcription factors (Liu & Dong, 2017). A large majority of ARF proteins contain three conserved domains, including a N-terminal B3 DNA binding domain (DBD), a middle region transcriptional activation domain (AD) or repression domain (RD), and a carboxy-terminal Aux/IAA dimerization domain (CTD) (Guilfoyle & Hagen, 2007; Huang et al., 2019).

As whole plant genomic sequences have been reported continuously, ARF gene families in many plant species have been systematically analyzed, such as 23 ARF genes in Arabidopsis thaliana (Okushima et al., 2005), 31 ARF genes in maize (Zea mays L.) (Xing et al., 2011), 25 ARF genes in rice (Oryza sativa L.) (Wang et al., 2007), 4 ARF genes in millet (Setaria italica L.) (Zhao et al., 2016), and 20 ARF genes in barley (Hordeum vulgare L.) (Huseyin, 2018). These data will significantly promote the functional studies of plant ARF genes.

In recent years, a large number of ARF genes have been cloned in plants and some of their functions have been studied. A. thaliana ARF5 (AtARF5) is the first plant ARF gene isolated by map-based cloning, and it plays an important role in the formation of embryo pattern and vascular tissue (Hardtke & Berleth, 1998). Mutations in AtARF1 and AtARF2 affect the growth patterns of pistils, as well as leaf senescence, floral organ abscission (Ellis et al., 2005). AtARF3 and AtARF4 play important roles in plant reproductive and nutritional growth (Pekker, Alvarez & Eshed, 2005). AtARF7 and AtARF19 promote lateral root formation and play important roles in hormone signaling pathway (Okushima et al., 2005; Feng et al., 2012). Transgenic rice (Oryza sativa L.) lines decreasing O. sativa ARF1 (OsARF1) expression are low vigor, stunt growth, have short curled leaves and are sterility, which suggests that OsARF1 plays an important role in both vegetative and reproductive organ developments (Attia et al., 2019).

Tillering ability is an important agronomic trait for grain production, and tiller bud outgrowth is an important factor determining tiller number (Li et al., 2003). Tiller bud growth is regulated by both genetic and environmental factors, and plant hormones are the direct regulators of both genetic and environmental factors (Zhang & Ma, 2015). The endogenous hormone indole acetic acid (IAA) is indirectly involved in the regulation of tiller bud growth (Choi et al., 2013), IAA is mainly synthesized in the shoot tip and young leaves, and it inhibits tiller bud growth by participating in the apical dominance, thus controlling the tiller occurrence (Ljung, Bhalerao & Sandberg, 2001). ARFs regulate the expression of auxin response genes (Guilfoyle & Hagen, 2007). Current study found that OsmiR167a repressed its targets, OsARF12, OsARF17 and OsARF25, to control rice tiller angle by fine-tuning auxin asymmetric distribution in shoots (Li et al., 2020). The transgenic rice plants overexpressing miR167 resulted in a substantial decrease the mRNA amount of four OsARF genes, OsARF6, OsARF12, OsARF17 and OsARF25, remarkably reduced tiller number (Liu et al., 2012).

At present, there are few studies on the regulation of wheat tillering by ARF genes. Research on mutant dmc helped us to confirm the importance of miR396b-TaARF11 in regulating tiller development (He et al., 2018). Besides, subsequent experiments showed that the contents of IAA in Guomai 301 and dmc were significantly different (An et al., 2019). In this study, all the ARF family members were identified using the version of wheat reference genome (RefSeq-v1.1) (IWGSC, 2018), and their evolution was studied. We thoroughly investigated the expression profiles of TaARF genes in Guomai301 and mutant dmc under normal growth and development condition, and exogenous IAA treatment. Also, we measured the endogenous hormone contents and analyzed the correlation between IAA and tiller capacity. These results provided a theoretical base for further research on the functions of ARFs in wheat.

Materials & Methods

Plant materials

Guomai 301 is a representative semi-winter wheat cultivar in Henan, China. It has dark green leaves, thick stems, long awns, large spindle-shaped spikes, and an average 37.4 grains per spike. These data were collected as described in previous study (Li et al., 2019).

Mutant dmc was obtained from EMS (ethyl methyl sulfonate) treated Guomai 301. The mutant and Guomai 301 were planted in our experimental field. Field management refers to conventional method (Li et al., 2014).

Tiller sample preparation and transcriptome sequencing

Three bulks of tiller samples were prepared separately at the three - leaf stage (WT1, dmc 1; sampling date: November 15th 2018), the over-winter stage (WT2, dmc 2; sampling date: January 6th 2019) and the rising to jointing stage (WT3, dmc 3; sampling date: February 16th 2019) for RNA extraction and used for qRT-PCR analysis. Wheat tillering had been completed at the rising to jointing stage.

Tiller primordia of Guomai 301 and mutant dmc at the three-leaf stage were dissected to carry out transcriptome sequencing (Fig. 1E). The tiller primordia at the three-leaf stage were carried out RNA-seq. The mutant dmc (T01, T02, and T03) and WT (T04, T05, and T06) had three biological replicates, respectively. The transcript abundance of TaARFs was calculated as fragments per kilobase of exon model per million mapped reads (FPKM) (Florea, Song & Salzberg, 2013). Differentially expressed genes (DEGs) between two sample pairs were analyzed using the DESeq R package (Wang et al., 2009). The false discovery rate (FDR <0.01) and fold change (FC ≥ 2) were set as the thresholds for DEGs. All analyses were performed on BMKCloud (https://www.biocloud.net/). The bioproject accession of the transcriptome data in NCBI is PRJNA670838. These data were collected as described in previous study (Li et al., 2019).

Figure 1 The tiller micromorphology of Guomai 301 (left) and mutant dmc (right).

(A) The individual plants of Guomai 301 and mutant dmc in the field condition. (B) The seedlings of Guomai 301 and dmc at the three-leaf stage. (C) The seedlings of Guomai 301 and dmc at the over-winter stage; (D) The seedlings of Guomai 301 and dmc at the rising to jointing stage. (E) Tiller primordia of Guomai 301 and dmc at the three-leaf stage. (F) Tiller primordia of Guomai 301 and dmc at the over-winter stage. (G) Tiller primordia of Guomai 301 and dmc at the rising to jointing stage. MC: main culm; TP: tiller primordium; Scale bar: 10 cm (A); 2 cm (B–D); 1 cm (E–G).

Determination of endogenous hormone contents

The tiller samples were prepared separately at the three-leaf stage (T1), the five-leaf stage (T2) and the over-winter stage (T3) for determination of endogenous hormone contents. IAA contents were extracted using a high-performance liquid chromatography method (Fang et al., 1998). Absorbance in each well was measured at 254 nm using a microplate reader (Thermo Scientific C18, Thermofisher, America). The samples at each stage had three independent replicates.

Continuous treatment of exogenous IAA and the tiller number record

The IAA solution is diluted with distilled water. Data were collected as previously described (Zhang et al., 2021). Specifically, the wheat seedlings of WT and mutant dmc at the two-leaf stage were sprayed with 10 µM IAA solution on the leaves until all the leaves were wet, and the controls were sprayed with distilled water. Each seedling was sprayed with 5 mL of water (control) or 10 µM IAA solution. The samples were treated once every three days for a total of 10 times. From the sixth time, the tiller numbers of the plants in different treatments were obviously different. After then, the tiller numbers of the plants were counted every 7 days. The results were analyzed using Excel for Microsoft Office 2016 according to average number.

Identification and characterization of TaARFs

Data were collected as previously described (Zhang et al., 2021). Specifically, the genome assembly version IWGSC refseqv1.1 (http://plants.ensembl.org/) was used to identify wheat ARF family. Considering that each gene in the wheat genome might have multiple transcripts, amino acid sequence corresponding to the longest transcript was used to identify ARF gene. The prediction of ARF proteins from the wheat genome were screened using the Hidden Markov Model (HMM). The HMM files corresponding to the B3 domain (PF02362) and auxin response domain (PF06507) were downloaded from the Pfam database (http://pfam.xfam.org/). HMMER 3.3 (http://www.hmmer.org/) (Finn, Clements & Eddy, 2011) was used to search the ARF genes from wheat genome database. All output protein sequences with e-value ≤ 1e−10 were collected. Additionally, keywords ‘ARF’ and ‘auxin response factor’ were employed to search against the Uniprotein database (https://www.uniprot.org/).

After removing all of the redundant sequences, the output putative ARF protein sequences were confirmed by CDD (https://www.ncbi.nlm.nih.gov/Structure/bwrpsb/bwrpsb.cgi), SMART (http://smart.embl-heidelberg.de/) and Pfam (http://pfam.xfam.org/) searching for the presence of the B3 domain and auxin response domain. Finally, obtained TaARFs were mainly referred to the annotation information from the Uniprotein database (https://www.uniprot.org/).

Protein and gene structures, chromosomal locations of ARF genes

The motif distribution was conducted using the MEME online tool (http://meme-suite.org/tools/meme). Parameters were set as following: the motif discovery mode was classic mode, the site distribution was Zero or One Occurrence Per Sequence (zoops), the maximum number of motif finding was 8, and other parameters were default. For exon-intron structure analysis, the DNA and cDNA sequences corresponding to each predicted protein from the wheat genome database were downloaded. The chromosomal map showed the physical locations of all identified ARF genes. All images were drawn using TBtools software (Chen et al., 2020). The prediction of isoelectric point (pI) and molecular weight (mw) of ARF genes were obtained from the ExPASy Proteomics Server (https://web.expasy.org/compute_pi/).

Analysis of the cis-acting elements in TaARF promoters

The 2000 bp upstream sequences of transcription start positions of TaARFs were extracted to carry out the analysis of cis-acting elements. The analysis was completed using the Plant CARE database (http://bioinformatics.psb.ugent.be/webtools/plantcare/html).

Chromosomal distribution and gene duplication

All ARF genes were mapped to wheat chromosomes based on physical locations information from the database of wheat genome using Circos (Krzywinski et al., 2009). Multiple Collinearity Scan toolkit (MCScanX) was adopted to analyze the gene duplication events, with the default parameters (Wang et al., 2012). Non-synonymous (ka) and synonymous (ks) substitution of each duplicated ARF gene were calculated using KaKs Calculator 2.0 (Wang et al., 2010). The syntenic maps were drawn using the Multiple Systeny Plot software (https://github.com/CJ-Chen/TBtools).

Phylogenetic analysis and classification of wheat ARF genes

A total of 23 ARF genes in Arabidopsis were obtained from TAIR database (https://www.arabidopsis.org/). A total of 25 ARF genes in rice and 31 ARF genes in maize were obtained from the Uniprotein database (https://www.uniprot.org/). The phylogenetic trees of the four species’ ARF genes were drawn using Neighbor-Joining (NJ) method of MEGA7.0 (http://www.megasoftware.net/), with the following parameters: Poisson model, pairwise deletion, and 1,000 bootstrap replications.

Analysis of ARF gene expression in various organs or tissues in wheat

The raw gene expression data were downloaded from the Wheat Expression Browser (http://www.wheat-expression.com/). A total of 13 RNA-sequencing data from wheat cultivar Chinese Spring were analyzed. These data were prepared from 13 tissues, including seeding, root, stem, flag leaf, spike, spikelet, awn, glume, lemma, anther, grain, stamen and pistil. Gene expression levels were estimated by the transcripts per million (TPM) values, and presented as log2-transformed normalized TPM. The heat map was drawn by TBtools software.

IAA treatment for gene expression analysis

The seeds of Guomai 301 and dmc were set in petri dishes for germination. After three days, the germinated seeds were planted in pot with soil and placed in a growth chamber at 23 °C and 50% relative humidity (RH), the light cycle was 16 h of light and 8 h of dark. The wheat seedlings at the early three-leaf stage were sprayed with distilled water, 1 × 10−5 mol/L IAA solution on the leaves, respectively. IAA was diluted with distilled water. The spray was completed until all the leaves were wet.

The tiller primordia of the seedlings sprayed with distilled water were sampled immediately and regarded as a control. The tiller primordia of the seedlings sprayed with IAA solution were sampled at 1 h and 2 h after treatments. All tiller primordia were dissected out with an anatomical needle after the out leaves and sheaths of seedlings were removed. The RNA samples of all treated tissues were immediately extracted and performed subsequent experiments.

qRT-PCR

Real time qRT-PCR was carried out as described in previous study (Li et al., 2019). Since the homoeoalleles of most tri-genes exhibited similar expression levels (Pfeifer et al., 2014), we used universal primers to analyze the expressions of TaARF homoeoallele genes. A total of 20 pairs of primers were designed based on the consensus sequences of homoeoalleles for every wheat ARF member, and the primers were listed in Table S1. The β-actin gene was used as an internal control and each reaction was performed with three biological replicates. The relative expressions of TaARFs were calculated by 2−ΔΔCT methods (Livak & Schmittgen, 2001).

Statistic analysis

All data were statistically analyzed. Values shown in the form of means ± SD were from three independent experiments. An asterisk (*) and two asterisks (**) indicate significant difference (P < 0.05) and highly significant difference (P < 0.01) using Student’s t-tests, respectively.

Results

Typical traits of Guomai 301 and mutant dmc

The mutant dmc (Fig. 1A) was mutagenized from wheat cultivar Guomai 301 (Fig. 1A). Mutant dmc almost didn’t tiller, and only had a main stem, and the plant height of the mutant dmc was significantly lower than that of the WT. At the three-leaf stage (Figs. 1B, 1E), two small tillers grew out at the base of the main culm in WT. Meanwhile, only one tiny protuberance formed at the main culm base of dmc. At the over-winter stage (Figs. 1C, 1F), the tiller number of WT was more than 6, while there were only two tiny tiller primordia (TPs) at the base of the dmc. Between the rising stage and the jointing stage (Fig. 1D), the tiny TPs of dmc were almost unchanged as before (Fig. 1J); but the tiller number of WT had reached its maximum value (Fig. 1G) (An et al., 2019).

The content change of endogenous IAA during wheat tiller formation

The IAA contents in dmc were significantly less than that in Guomai 301 at the three-leaf stage and the five-leaf stage (Fig. 2), and the IAA contents in Guomai 301 were 1.6-fold and 1.3-fold of that in dmc, respectively. While the IAA content in Guomai 301 was significantly less than that in dmc at the over-winter stage, the content of IAA in dmc was 5.4-fold of that in Guomai 301. Besides, the contents of IAA in Guomai 301 and dmc were increased at the five-leaf stage and decreased at the over-winter stage, indicating IAA played essential roles in wheat tiller growth and development.

Figure 2 The endogenous IAA contents in tiller primordia of Guomai301 and dmc.

S1: the three-leaf stage; S2: the five-leaf stage; S3: the over-winter stage. Asterisks indicate significant difference or highly significant difference between Guomai 301 and dmc in different stages.

Effects of exogenous IAA on wheat tiller formation

On the 18th day after IAA treatment (Fig. 3, T1), the tiller number of Guomai 301 was significantly increased, while dmc and the control of Guomai 301 remained no tiller. The exogenous IAA continuously promoted the tillering of Guomai 301, but the effect was less on dmc (Fig. 3). The data indicated that exogenous IAA could significantly promoted tiller development of Guomai 301, but it had less effect on dmc, which suggested that dmc was insensitive to IAA.

Figure 3 The tiller number changes of Guomai 301 and dmc in response to IAA treatments.

T1-T6 of the x-axis indicated the sampling dates, and the tiller numbers were recorded every 7 days. T1 is the first sampling date which was the 18th day after IAA treatment. Asterisks indicate significant difference or highly significant difference between treated groups and control groups in different sampling dates, respectively.

Genome wide discovery of wheat ARFs

A total of 74 candidate ARFs were initially obtained from all wheat protein sequences using HMM (PF02362 and PF06507) by HMMER3.3. The validation of protein conserved domains showed that seven sequences hadn’t AUX_IAA or Auxin_resp domains, which indicated that the seven sequences were not typical ARFs. Eventually, we obtained a total of 67 unique ARF genes in wheat. Detailed information about each ARF gene was showed in Table S2.

Among the 67 ARF proteins, TaARF13-7D was identified as the smallest protein with 354 amino acids (aa), whereas the largest one was TaARF19-7D with 1175 aa. The molecular weight of the proteins ranged from 38829.72 Da (TaARF13-7D) to 130932.17 Da (TaARF19-7D), and the theoretical pI ranged from 5.42 (TaARF13-2D) to 8.7 (TaARF3-3D).

Phylogenetic tree of the wheat ARF proteins

An unrooted phylogenetic tree was generated by using the amino acid sequences of a total of 146 ARF proteins from four species (Fig. 4). The result clearly clarified the phylogenetic relationships among the ARFs. According to the bootstrap value of the phylogenetic tree, these ARFs were clustered into two classes (Class I and Class II), including four subfamilies (Ia, Ib, IIa, IIb). Among them, the Class II contained more ARF proteins.

Figure 4 Phylogenetic tree of ARF proteins from Arabidopsis, maize, rice and wheat.

The purple solid diamonds represent ARF proteins in Arabidopsis (AtARF); The green squares represent ARF proteins in maize (ZmARF); The blue deltas represent ARF proteins in rice (OsARF); The red solid circles represent ARF proteins in wheat (TaARF); The different colored sectors indicate different groups (or subgroups) of ARF proteins. The different colored arcs indicate different classes of ARF proteins.

Clustering of protein sequences from different species indicated that the ARFs in the same subfamily were highly similar, which implied their similar functions and evolution processes. Compared to Arabidopsis, wheat ARFs were more closely related to those of maize and rice.

Motif pattern, domain pattern of wheat ARF proteins

To better understand the structural characteristics of ARF proteins in each subfamily, ten conserved motifs were identified in ARF proteins using MEME motif search tool (Fig. 5B, Table S3). Only TaARF13-7D had the least number of motif modules. Motif 1, motif 3 and motif 4 modules were shared by all ARF proteins. Motif 1, motif 2, motif 3, motif 4, motif 5, motif 6, motif 7, motif 9 and motif 10 modules were shared by Class II. Typically, motif 8 existed in subfamily IIb, but without in subfamily IIa.

Figure 5 Phylogenetic relationships, conserved protein motif patterns, domain patterns and gene structures of TaARFs.

(A) The phylogenetic tree of TaARF proteins. Clusters are indicated with different colors. (B) The motif compositions of TaARFs. The 1–10 motifs are displayed in different colored boxes, the scale at the bottom indicates the length of proteins. (C) The domain patterns of TaARFs, the B3 domains are highlighted in yellow, the auxin response domains are highlighted in green, and the AUX_IAA domain are highlighted in lilac. (D) Exon-intron structures of TaARFs, yellow boxes indicate 5′- and 3′- untranslated regions; green boxes indicate exons; black lines indicate introns.

According to the result of the domain prediction, the proteins in Class I subfamily had B3 and auxin response domains. The proteins in Class II subfamily had B3, auxin response and AUX_IAA domains (Fig. 5C).

Gene structure of TaARFs

The exon-intron organizations of all the identified TaARFs were visualized (Fig. 5D). TaARFs possessed two to fourteen exons. Genes within the same group usually had similar structures. For example, all ARF genes in Class II contained thirteen exons and fourteen introns, and all ARF genes in Class Ia contained three exons and two introns. Among them, TaARF13-2A, TaARF13-2D, TaARF13-7A and TaARF13-7D had only two exons.

Cis-acting elements in the promoters of TaARFs

Among the TaARFs, the promoter sequences of 17 TaARF genes contained a large number of ‘N’, so they hadn’t been analyzed (Fig. 6). CAT-box and CCGTCC motif cis-elements related to growth development exist commonly in the promoter sequences of TaARFs. In addition, there are also a large number of hormone response-related cis-elements, including some cis-acting elements involving in auxin (AuxRR-core, TGA-element), gibberellin (P-box), methyl jasmonate reaction (CGTCA-motif), salicylic acid response (TCA-element), abscisic acid response (ABRE) and ethylene response (ERE). AuxRR-core or TGA-element is the most cis-elements, 32 TaARFs contain AuxRR-core or TGA-element. Each TaARF contains at least two cis-elements. For example, TaARF25-5D has a growth-related cis-element (CAT-box) and a hormone response-related cis-element (ABRE). These cis-acting elements implied TaARFs play various roles in regulating wheat growth and development, and respond to multiple hormones.

Figure 6 The cis-acting elements in the promoters of TaARFs.

Growth-related cis-element: meristem expression regulation (CAT-box and CCGTCC motifs); hormone response-related cis-elements: abscisic acid response (ABRE), methyl jasmonate response (CGTCA-motif), salicylic acid response (TCA-element), gibberellic response (P-box), auxin response (TGA-element and AuxRR-core) and ethylene response (ERE).

Chromosomal localizations and duplications of TaARF genes

The 67 TaARFs were distributed on 18 wheat chromosomes randomly. The majority of TaARFs were located on the distal ends of the chromosomes. Chromosome 7A contained the largest number of ARF genes (7). No ARF gene was identified on the homoeologous chromosomes 4A, 4B and 4D, and only one ARF gene and its homoeologous genes (TaARF25-5A, TaARF25-5B and TaARF25-5D) were located on homoeologous chromosomes 5A, 5B and 5D. Four pairs of tandem duplicated genes (TaARF4-3A-1 and TaARF4-3A-2, TaARF4-3B-1 and TaARF4-3B-2, TaARF4-3D-1 and TaARF4-3D-2, TaARF13-7A-1 and TaARF13-7A-2) were located on 3A, 3B, 3D and 7A, respectively. Chromosome 2D, 3A, 3B, 3D and 7A, 7B, 7D had the most ARF genes.

The tandem duplication events (Fig. 7) involving chromosomal localizations of ARF genes were used to directly discover the distribution of the duplication of ARF genes in the wheat genome. 89 segmental duplication events among 67 ARF genes were identified (Table S4). In other words, all TaARFs were involved in chromosome segmental duplication. Most TaARFs were associated with two to three syntenic gene pairs. Some TaARFs had at least three syntenic gene pairs on the same chromosome, such as TaARF3-3A, TaARF3-3B, TaARF3-3D, TaARF15-1A, TaARF15-1B and TaARF15-1D.

Figure 7 Schematic diagram of the chromosome distribution and interchromosome relationships of TaARFs.

The grey lines indicate all duplicated gene pairs in wheat, the highlighted red lines indicate probably duplicated TaARF gene pairs.

Not only chromosome segmental duplication events occurred on the same chromosome, but also occurred between different chromosomes. For example, chromosome 1 and chromosome 3, chromosome 2 and chromosome 6, a total of 19 chromosome segmental duplication events were discovered. These results indicated that the chromosome segmental duplication was a major driving force for TaARF evolution.

Evolutionary relationships of ARF genes in wheat and three different species

In order to further understand the evolution mechanism of ARF genes among different species. Three comparative syntenic maps associated with wheat genome were constructed with Arabidopsis, rice and maize genomes (Fig. 8). The numbers of the orthologous ARF gene pairs between wheat and the three species (Arabidopsis, rice and maize) were 6, 98 and 105, respectively (Table S5).

Figure 8 Syntenic relationships of ARF genes between wheat and three representative species.

Gray lines in the background indicate the collinear blocks within wheat and other plant genomes, while the blue lines highlight the syntenic ARF gene pairs.

Six of the 67 TaARFs (TaARF22-1A, TaARF22-1B, TaARF22-1D, TaARF16-7A, TaARF16-7B and TaARF16-7D) had syntenic relationship with two Arabidopsis ARF genes (AtARF10 and AtARF7) (Fig. 8A). TaARFs had higher syntenic relationship with grass plants rice and maize (Figs. 8B, 8C). 58 TaARFs (including 21 TaARFs and their homoeologous genes) had syntenic relationship with 19 maize ARF genes (Fig. 8B), 59 TaARFs (including 21 TaARFs and their homoeologous genes) had syntenic relationship with 22 rice ARF genes (Fig. 8C). Especially, the syntenic gene of wheat TaARF5-6D was identified in rice, but not in maize.

The Ka/Ks ratios of the ARF gene pairs between wheat and other species (Table S5) showed that all segmental and tandem duplicated gene pairs had Ka/Ks <1, suggesting the TaARF genes might have experienced strong purifying selective pressure during evolution. In addition, the ARF genes in grass plants of wheat, rice and maize were highly conserved in the syntenic blocks, for they had a closer phylogenetic relationship, and these TaARFs were evolved from ancient ARF orthologous genes.

The expression patterns of TaARFs in different tissues

The expression profiles of all the 67 TaARFs during development were analyzed with the transcriptome data from the Wheat Expression Browser (http://www.wheat-expression.com/), which were derived from 13 wheat organs/tissues at different developmental stages (Fig. 9A). There were four typical expression profiles. (1) TaARFs expressed very lowly in all tissues during wheat development, such as TaARF2, TaARF8, TaARF11 and TaARF13. (2) TaARFs expressed highly in all tissues during wheat development, such as TaARF4 and TaARF9. They probably play basic important roles during wheat development. (3) TaARFs expressed in all tissues during wheat development, but the expression levels were relative lower, such as TaARF3 and TaARF22. They probably also play basic roles during wheat development. (4) TaARFs expressed highly only in specific tissues or their expression levels were changed during wheat development, such as TaARF17 expressed highly in stem and TaARF22 expressed highly in spikelet. Most TaARFs belong to this class and they play vital roles in various organ developments. Most TaARF homoeologous genes had similar expression patterns. Four pairs of tandem duplicated genes (TaARF4-3A-1 and TaARF4-3A-2, TaARF4-3B-1 and TaARF4-3B-2, TaARF4-3D-1 and TaARF4-3D-2, TaARF13-7A-1 and TaARF13-7A-2) showed remarkably different expression profiles, suggesting they evolved from different orthologous genes.

Figure 9 Expression profiles of TaARFs in various organs or tissues.

(A) Heatmap of expression profiles of TaARFs in various organs or tissues of Chinese Spring from the Wheat Expression Browser (http://www.wheat-expression.com/). (B) The heat map of expression profiles of TaARFs in tiller primordia of WT and dmc based on transcriptome data. Three biological replicates were set up in the mutant dmc (T01, T02 and T03) and WT (T04, T05 and T06), and each sample bulk of tiller primordia included more than 10 independent individuals.

The expression profiles of TaARFs in tiller primordia showed that the transcripts of five TaARF genes (TaARF1-3A, TaARF1-3B, TaARF13-7A-2, TaARF14-1B and TaARF19-7D) had not been detected in WT and mutant dmc, which indicated their very lower expression levels (Fig. 9B). TaARF4, TaARF9, TaARF12, TaARF15, TaARF17, TaARF21, TaARF25 and their homoeologous genes had higher expression levels (FPKM>10), but their expressions were not significant differences between WT and mutant dmc. High expression levels suggested they played basic important roles during tiller development. In addition, compared to WT, most TaARF genes showed low expression levels in mutant dmc. Only 4 TaARF genes (TaARF2-3D, TaARF11-2A, TaARF11-2B and TaARF11-2D) expressed differentially between WT and dmc (FC > 2), and they all expressed lowly in mutant dmc. Most TaARFs expressed relatively lower at early tillering stage in mutant dmc, this should be a major factor constraining tillering of the dmc.

In summary, TaARF3, TaARF4, TaARF9 and TaARF22 and their homoeologous genes played basic roles during wheat development. TaARF4, TaARF9, TaARF12, TaARF15, TaARF17, TaARF21, TaARF25 and their homoeologous genes probably play basic important roles during tiller development.

Expression profiles of TaARFs in tiller primordia of the mutant dmc

According to the transcriptomics data, most TaARF genes showed no significant differential expressions (FC < 2) at the three-leaf stage. qRT-PCR was performed to analyze the expression patterns of 20 TaARFs in the tiller primordia of WT and mutant dmc at three tiller developmental stages (Fig. 10), and the samples at the three-leaf stage (WT1 and dmc 1) were consistent with the samples of RNA-sequencing. The 20 TaARF genes had various expression patterns at three tillering stages.

Figure 10 QRT-PCR results of 20 TaARFs in the tiller primordia of WT and dmc at three tillering stages.

WT1, dmc 1: the three-leaf stage; WT2, dmc 2: the over-winter stage; WT3, dmc 3: the rising to jointing stage. Data were normalized to β-actin gene and vertical bars indicated standard deviation. Asterisks indicate significant difference or highly significant difference between Guomai 301 and dmc.

Among them, TaARF2, TaARF3, TaARF13-2A, TaARF16 and TaARF19 showed no significant differential expressions at three tillering stages, and most TaARF genes showed no significant differential expressions at the over-winter stage, except for TaARF11 and TaARF17. At the rising to jointing stage, TaARF4, TaARF5, TaARF10, TaARF11, TaARF12, TaARF14, TaARF18 and TaARF22 had higher expression levels in mutant dmc. A total of 4 TaARF genes showed significant differential expression levels between WT and dmc at the three-leaf stage, including TaARF11, TaARF13-7A, TaARF14 and TaARF17. More importantly, these 4 TaARF genes were all down-regulated in mutant dmc. It indicated that only a few key genes exerted a significant effect on tiller formation at three leaf stage, the constrained tillering of the dmc was associated with the lower expression levels of TaARFs. Besides, TaARF11 and TaARF14 had similar expression patterns, and they expressed lowly in dmc at the over-winter stage but expressed highly at the rising to jointing stage.

In summary, the expression patterns of TaARF genes were complex. The abnormal expressions of TaARF11 and TaARF14 were major causes in constraining the tillering of dmc.

Expression patterns of TaARFs in response to IAA

The cis-acting element analysis showed that a number of hormone response-related cis-elements existed in the promoter regions of TaARF genes. Typically, cis-acting elements involved in auxin regulation. 20 TaARF genes were investigated whether their expressions were affected by IAA treatment (Fig. 11).

Figure 11 Expression profiles of 20 TaARFs in response to IAA treatment.

Data were normalized to β-actin gene and vertical bars indicated standard deviation. Asterisks indicate significant difference or highly significant difference between Guomai 301 and dmc.

The expressions of 6 TaARF genes (TaARF2, TaARF4, TaARF5, TaARF8, TaARF13-2A and TaARF15) were significantly up-regulated in mutant dmc at 1 h after IAA treatment, among the six TaARF genes, three TaARF genes (TaARF4, TaARF5 and TaARF8) were significantly down-regulated in mutant dmc at 2 h after IAA treatment. Compare to mutant dmc, the expression levels of 7 TaARFs (TaARF9, TaARF11, TaARF13-2A, TaARF15, TaARF17 and TaARF21) in WT were continuously repressed by IAA treatment, especially, the expression levels of TaARF15 and TaARF13-7A decreased by more than 50% at 1 h and 2 h after IAA treatment. TaARF13-7A had the most TGA-element (3) (Fig. 6). The promoter region of TaARF15 contained a large number of ‘N’, so it was not analyzed. It was speculated that the auxin-related cis-acting elements determined the expressions of TaARFs response to IAA stimulating. The expressions of TaARF3 changed not significantly, which suggested it was not sensitive to IAA stimulation. Contrarily, the expressions of TaARF8 and TaARF15 were significantly affected by IAA in WT and dmc, which suggested they were sensitive to IAA stimulation and might play key roles in regulating wheat tillering.

Discussion

Characteristics and evolution of TaARFs

Up to now, ARF gene families have been identified in various species, including wheat. In this study, the use of multiple identification methods at the same time greatly improved the accuracy of the wheat ARF genes. A total of 23 wheat ARF members encoded by 68 homoeoalleles are identified from wheat reference genome version TGACv1 (Qiao et al., 2018), and 61 TaARF genes are identified from genome version IWGSC1+ popseq.31 (Sun et al., 2018). In this study, 67 TaARF genes, including 21 homoeologous TaARF loci, distributed on 18 chromosomes were identified in wheat using the latest version of wheat reference genome (RefSeq-v1.1) (IWGSC, 2018), which was the best version of wheat chromosome scale assembly now. The annotation of each TaARF gene was carried out referred to the Uniprotein database (https://www.uniprot.org/). Unified annotation will help these results have a wider applicability to the broader field. All these TaARFs were highly conserved, and encoded proteins with typical domains of plant ARFs.

Wheat derives from a grass ancestor structured in seven protochromosomes followed by a paleotetraploidization to reach a 12 chromosomes intermediate and a neohexaploidization (involving subgenomes A, B and D) event that finally shaped the 21 modern chromosomes (Pont et al., 2013). Because wheat is a heterohexaploid plant species, it has more ARF genes than Arabidopsis (23) (Okushima et al., 2005), rice (25) (Wang et al., 2007) and maize (31) (Xing et al., 2011). The loss of ARF genes on chromosome 4 (4A, 4B and 4D) might result from recombinant or modification of some redundant genes during wheat evolution (Chen, 2007; Otto, 2007). Most TaARF genes in the same subfamily have similar exon/intron structures, which provide clues to the evolutionary relationships of TaARFs (Hu & Liu, 2011). These data indicate that the ARF genes with similar structures have similar evolution histories and functions (Babenko et al., 2004; Roy & Penny, 2007). A large number of cis-acting elements related to growth and development and hormones regulation existed in the promoter regions of TaARF genes, which implied their various functions. TaARFs had a poor collinearity with ARFs of Arabidopsis, but had a better collinearity with ARFs of rice and maize. All TaARFs might have happened segmental duplication, which had played a fundamentally important role in TaARF evolution (Zhang, 2000; Leister, 2004).

The protein sequences and gene structures of homoeologous genes TaARF4-3A-1, TaARF4-3B-1, and TaARF4-3D-1 were highly similar, and that of homoeologous genes TaARF4-3A-2, TaARF4-3B-2, and TaARF4-3D-2 were highly similar (Fig. 4, 5, 6), so we concluded that the two homoeologous genes evolved parallel from wheat species formation. The expression profiles of TaARF4-3A-1, TaARF4-3B-1, and TaARF4-3D-1 were similar, but were significantly different from that of TaARF4-3A-2, TaARF4-3B-2, and TaARF4-3D-2, which demonstrated the conclusion (Fig. 9). The protein and promoter sequences, and gene structures of TaARF13-7A-1 and TaARF13-7A-2 were almost the same, which indicated they were duplicated genes happened not long before (Fig. 6). Except for TaARF13-7A-1 and TaARF13-7A-2 were duplicated genes happened recently, the most TaARFs were evolved parallel from wheat species formation.

Various functions of TaARFs

Gene structural similarity determines its functional similarity. Plant ARF genes in the same subfamily have similar functions (Fig. 4). For example, disruption and overexpression of AtARF8 affect hypocotyl elongation and root growth habit (Tian et al., 2004). Transgenic experiments show that the ARF8 can promote or inhibit lateral root formation in Arabidopsis (Yang et al., 2006). AtARF4 plays an important role in the reproductive and nutritional growths (Pekker, Alvarez & Eshed, 2005). Similarly, TaARF4 determines root length and plant height in wheat (Wang et al., 2019). These results indicated that homologous ARF genes from different plant species might have similar functions. Most TaARF homoeologous genes in A, B and D genomes exhibited similar spatiotemporal expression profiles (Pfeifer et al., 2014), such as TaARF 1/4/9/12/15/17/21/25 and their homoeologous genes (Fig. 9). This data also suggested the homoeologous TaARFs had similar functions.

Most ARF genes have different tissue-specific expression patterns, suggesting their special functions in different tissue/organ development. For example, ARF7 and ARF19 regulate lateral root formation in Arabidopsis (Fukaki, Taniguchi & Tasaka, 2006; Okushima et al., 2007). Transgenic Arabidopsis lines expressing TaARF15-A.1 promotes the growth of roots and leaves (Qiao et al., 2018). OsARF19 is pivotal for floral organ development and plant architecture (Zhang et al., 2015). ARF17 is essential for pollen wall patterning in Arabidopsis by modulating primexine formation at least partially through direct regulation of CalS5 gene expression (Yang et al., 2013), and the overexpression of ARF17 in the tapetum and microsporocytes of 5mARF17/WT plants leads to male sterility (Wang et al., 2017). Overexpression of AtTTP affects ARF17 expression and leads to male sterility in Arabidopsis (Shi et al., 2015). Up to now, most functional studies of ARF genes have been carried out in A. thaliana. Most TaARFs also have typical tissue-specific expression profiles (Fig. 9), which suggests their various functions in wheat development.

The key TaARFs involved in tiller development

Plant ARF genes play an important role in maintaining plant stem apical meristem (Zhao et al., 2010). The enhanced miR167 level in transgenic rice resulted in a substantial decrease in mRNA amounts of the four OsARF genes, OsARF6, OsARF12, OsARF17 and OsARF25, the transgenic rice plants remarkably reduced tiller number (Liu et al., 2012). Recent research suggested OsmiR167a could repress OsARF12, OsARF17 and OsARF25, to control rice tiller angle by fine-tuning auxin asymmetric distribution in shoots (Li et al., 2020). Our miRNome and transcriptome integrative analysis about the mutant dmc and WT found that the highly expressed tae-miR396b (T. aestivum microRNA396b) significantly repressed the expressions of TaGRF genes and TaARF11 in dmc during tillering (He et al., 2018). It was predicted that the miR396b/ARF11 regulatory module played a key role in wheat tiller development. Compared with the WT, the expressions of four TaARFs, TaARF11, TaARF13-7A, TaARF14 and TaARF17, in dmc were significantly decreased at early tillering stage, which was positively related to the phenotype of dmc (Fig. 10). Most TaARFs had different expression patterns in WT and dmc, but only those significantly differentially expressed TaARFs in tiller primordia were the key tiller development regulators. In this case, TaARF11 and TaARF14 were significantly differentially expressed at early tillering stage, indicating their important roles in regulating tiller numbers in wheat.

IAA affect the expressions of TaARFs and significantly promoted tillering

Hormone responses are fundamental to the development and plastic growth of plants (Chapman & Estelle, 2009). There are a number of evidences that exogenous IAA can obviously influence rice and wheat tillering (Kariali & Mohapatra, 2007; Liu et al., 2011; Assuero et al., 2012; Cai et al., 2013). Apically derived auxin does not enter axillary buds directly in several species, including in Arabidopsis (Booker, Chatfield & Leyser, 2003). Apical auxin can inhibit the growth of small buds, and it has been proposed that its inhibitory effect is mediated by a second messenger (Chatfield et al., 2000). In rice, there are many genes related to tiller number may also be related to various plant hormones, rice dwarf and low tillering 10 (OsDLT10) regulates tiller number by monitoring auxin homeostasis (Wen et al., 2020). The phytohormone auxin is involved in almost all developmental processes in land plants, different ARF genes probably contribute to the establishment of multiple unique auxin responses in plant development (Roosjen, Paque & Weijers, 2017). In our study, the TaARF genes showed various expression patterns after IAA treatment. There are a large number of cis-acting elements related to hormones in TaARF promoters, including those related to IAA (AuxRR-core, TGA-element). Tissue specific promoters control gene expression in certain organs or tissues (Li & Chen, 2015). The results of qRT-PCR also confirmed that the expressions of TaARFs were significantly affected by IAA treatment (Fig. 11). IAA contents in dmc were significantly less than that in Guomai 301 at key tillering stages (Fig. 2), and IAA application significantly promoted wheat tillering (Fig. 3). According to these data, it was considered that TaARFs as well as IAA signaling were involved in regulating wheat tiller development.

Conclusions

A total of 67 TaARFs were identified in wheat. TaARF genes distribute on 18 wheat chromosomes randomly, and their promoter regions have a large number of cis-acting elements related to plant growth and development, and hormone response. The most TaARFs evolved parallel from wheat formation, except for TaARF13-7A-1 and TaARF13-7A-2 duplicated recently. The homoeologous TaARFs are highly similar and also have similar expression profiles. TaARF3, TaARF4, TaARF9 and TaARF22 and their homoeologous genes play basic roles during wheat development. TaARF4, TaARF9, TaARF12, TaARF15, TaARF17, TaARF21, TaARF25 and their homoeologous genes play basic roles during tiller development. The abnormal expressions of TaARF11 and TaARF14 are major causes constraining the tillering of dmc. The IAA contents of dmc are significantly less than that in WT during key tillering stages. Exogenous IAA significantly affected the expressions of TaARFs and promoted wheat tillering, which demonstrated that TaARFs and IAA signaling were involved in controlling wheat tillering. This study provided valuable clues for functional characterization of ARFs in wheat.

Supplemental Information

Supplemental Information 1 DNA sequences of the primers used in this study

Click here for additional data file.

Supplemental Information 2 The basic information of ARFs in wheat

Click here for additional data file.

Supplemental Information 3 Conserved motif analysis of wheat ARF proteins

Click here for additional data file.

Supplemental Information 4 One-to-one orthologous relationships of ARFs between wheat and and other species

Click here for additional data file.

Supplemental Information 5 The duplication gene pairs of ARFs in wheat genome

Click here for additional data file.

Supplemental Information 6 Raw data for q-RT-PCR

Click here for additional data file.

We are grateful for the assistance by Shangqiu Academy of Agricultural and Forestry Sciences. We thank National Centre of Engineering and Technological Research of Wheat for the technical support for the cultivations.

Additional Information and Declarations

Competing Interests

Author Contributions

Data Availability

The authors declare there are no competing interests.

Junchang Li conceived and designed the experiments, performed the experiments, prepared figures and/or tables, authored or reviewed drafts of the paper, and approved the final draft.

Yumei Jiang, Jing Zhang, Yongjing Ni and Zhixin Jiao conceived and designed the experiments, authored or reviewed drafts of the paper, and approved the final draft.

Huijuan Li, Ting Wang, Peipei Zhang, Wenlong Guo performed the experiments, authored or reviewed drafts of the paper, and approved the final draft.

Lei Li analyzed the data, authored or reviewed drafts of the paper, and approved the final draft.

Hongjie Liu, Hairong Zhang, Qiaoyun Li and Jishan Niu analyzed the data, prepared figures and/or tables, and approved the final draft.

The following information was supplied regarding data availability:

The sequences are available at NCBI BioProject (PRJNA670838) and the raw measurements are available in the Supplemental File.

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
