# Peer review of "Key auxin response factor (ARF) genes constraining wheat tillering of mutant dmc"

_PeerJ, doi:10.7717/peerj.12221_

## Round 0.1 · original submission · Minor Revisions

We have received the reviewer comments from two experts in this field, both of them feel that it is an interesting study and the results obtained in this study is significant. However, the reviewers have also raised some concerns need to be addressed in the revised manuscript. Please read the reviewer comments carefully and addressed them seriously before resubmit your manuscript. In particular, the authors should pay attention to the statistical analysis, which was pointed out both reviewers. In addition, the authors should consider re-performing qRT-PCR with a second control gene as suggested by Reviewer 1. Furthermore, the authors should discuss more about the mechanisms underlying the action of auxin in controlling tillering in plants. We are looking forward to receiving your manuscript.

Reviewer 1 ·

Basic reporting

no comment

Experimental design

no comment

Validity of the findings

no comment

Additional comments

This manuscript identified 67 ARF genes encoding auxin response factor in wheat, putatively involved in IAA signaling to control wheat tillering. This study provided valuable clues for functional characterization of ARFs in wheat. However, some data must be re-organized and re-analyzed. There are also too much grammar errors in the manuscript, which must be improved in language.


Some comments:

1: In line 52, the “Oryza sativa L.” should be added between rice and (Wang et al., 2007) as the same as maize, millet and so on.

2: Statistical analyses are not described in the materials and methods. A section should be added to describe these. Subsequently, significant differences should be examined between WT and dmc in Figure2, Figure 10 and Figure 11.

3: I am confused why the author chose 10 μM IAA to spray the wheat seedlings (Figure 3)? Did the authors perform assay that would generate curve at the different concentrations?

4: Quantitative real-time PCR experiments normalisation is usually conducted with at least two validated reference genes. Please provide new experimental data with a second control gene to show stable expression of the first reference gene under all experimental conditions.

Reviewer 2 ·

Basic reporting

no comment

Experimental design

no comment

Validity of the findings

no comment

Additional comments

The authors conducted a systematic and comprehensive analysis of wheat ARF family genes, providing valuable clues for functional characterization of ARFs in wheat. Analysis of wheat ARF family genes suggest that TaARFs and IAA signaling were involved in controlling wheat tillering, and the abnormal expressions of TaARF11 and TaARF14 were major causes constraining the tillering of dmc mutants. This study provides a necessary foundation for wheat tillering regulation mechanism and for breeding by increasing tillers to increase yield.
1. Please add statistical results for qRT-PCR data in Figure 10 and Figure 11. For readers’ convenience, the legends should be more detailed, and key genes should be emphasized in order to get a clear conclusion directly from your figure.
2. Due to the mechanism of auxin in regulating tillering has been reported in Arabidopsis and rice, more discussion should be given about the function of auxin in tillering.
3. Please reconsider if it is necessary to emphasize dmc mutant in the title.

---

## Round 0.2 · Minor Revisions

I am happy to see that all concerns raised during the last round of review have been fully addressed. Before it is publishable in PeerJ, please address the following comments from the Section Editor:

> the authors should highlight the importance and better emphasize the applicability of their findings to the broader field. It would also be nice if the discussion 'discussed' many of the comparisons and cited literature referred to.

---

## Round 0.3 · accepted · Accept

Thanks for fully addressing the raised concerns. I feel that the quality of this revised manuscript has been improved substantially, and it is now publishable in PeerJ. Congratulations!